# A Chemical Damage Creep Model of Rock Considering the Influence of Triaxial Stress

**DOI:** 10.3390/ma15217590

**Published:** 2022-10-28

**Authors:** Youliang Chen, Qijian Chen, Yungui Pan, Peng Xiao, Xi Du, Suran Wang, Ning Zhang, Xiaojian Wu

**Affiliations:** 1Department of Civil Engineering, School of Environment and Architecture, University of Shanghai for Science and Technology, Shanghai 200093, China; 2Department of Engineering Geology and Hydrogeology, RWTH Aachen University, 52064 Aachen, Germany; 3School of Transportation, Southeast University, Nanjing 210096, China; 4School of Civil and Environmental Engineering, University of New South Wales, Sydney 2052, Australia; 5Shanghai Construction Engineering Group Engineering Research Institute, Shanghai 200093, China

**Keywords:** chemical corrosion, true triaxial stress, nonlinearity, rock creep constitutive, intermediate principal stress

## Abstract

In order to accurately describe the characteristics of each stage of rock creep behavior under the combined action of acid environment and true triaxial stress, based on damage mechanics, chemical damage is connected with elastic modulus; thus, the damage relations considering creep stress damage and chemical damage are obtained. The elastic body, nonlinear Kelvin body, linear Kelvin body, and viscoelastic–plastic body (Mogi–Coulomb) are connected in series, and the actual situation under the action of true triaxial stress is considered at the same time. Therefore, a damage creep constitutive model considering the coupling of rock acid corrosion and true triaxial stress is established. The parameters of the deduced model are identified and verified with the existing experimental research results. The yield surface equation of rock under true triaxial stress is obtained by data fitting, and the influence of intermediate principal stress on the creep model is discussed. The derived constitutive model can accurately describe the characteristics of each stage of true triaxial creep behavior of rock under acid environment.

## 1. Introduction

Rock rheology is the study of how rocks’ mechanical properties change with time, including rheological properties such as creep, stress lax, elastic aftereffect, etc., and is closely related to the long-term stability and safety of rock engineering. A large number of engineering practices and theoretical studies have shown [1,2,3] that the damage and instability found in rock engineering usually occur during the project operation, and the damage to property and loss of life caused by rock rheology are often very serious [4,5]. Therefore, studying the rheological constitutive model of rock has important theoretical significance for the stability and safety of practical engineering. In recent years, many staged results have been achieved in the derivation and establishment of rock creep constitutive models: researchers established various creep models including the empirical model, component combination model, and phenomenological model [6,7,8,9,10]. Hou et al. [11] obtained a four-element combination model considering the damage of sandstone which was well-verified according to the existing experimental data. Jiang et al. [12] obtained a Nishihara model and the correctness of the model was verified by the experimental data of sandstone, which reflected the characteristics of multiple stages of the creep process. Zhou et al. [13] improved the Nishihara model, considering the damage of rock, and accurately verified the model. Wang et al. [14] regarded the macroscopic parameters of rock as a function related to time, and derived a nonlinear model that could reflect the creep characteristics of rock. Liu et al. [15] carried out an indoor triaxial creep test on the deep surrounding rock from Fuxin Hengda Coal Mine, and analyzed the creep deformation law of the rock, and a constitutive model that could reflect the nonlinear accelerated creep stage was obtained and verified with experimental data. Zhang et al. [16] pointed out the inaccuracy of parameter identification in the process of establishing the creep model in previous studies, and established a model that could describe the instantaneous elastic strain, decay creep, isokinetic creep, and accelerated creep stages.

The rock creep models established by the above scholars were all obtained under uniaxial or conventional triaxial conditions, and the research on rock creep under the true triaxial stress state is still relatively lacking. In practical engineering, the rock is often in a state of three-phase stress, and the existence of intermediate principal stress will significantly affect the instantaneous and time-dependent properties of the rock [17,18].

At the same time, the environment in which rock formation occurs was often complex and changeable. For example, in the acid rain area of the Sichuan Basin in southwest China, the acidic environment of the rock will have an impact on the mechanical properties of rock engineering. The chemical composition contained in the acidic solution reacts with the rock, changing its microstructure and reducing its bearing capacity. Many studies have shown [19,20,21,22,23] that chemical solutions have a significant impact on the development of damage and the mechanical properties of rock, but these results are mostly aimed at the transient mechanical properties of rocks, including laboratory tests and models; furthermore, research on the creep characteristics of rocks under chemical corrosion is relatively lacking.

Therefore, in order to study the time-dependent characteristics of rock in acidic environment and true triaxial stress, based on damage mechanics, chemical damage is connected with elastic modulus; thus, the damage relations considering creep stress damage and chemical damage are obtained. The elastic body, nonlinear Kelvin body, linear Kelvin body, and viscoelastic–plastic body (Mogi–Coulomb) are connected in series, and the actual situation under the action of true triaxial stress is considered at the same time. Therefore, a damage creep constitutive model considering the coupling of rock acid corrosion and true triaxial stress is established. The parameters of the deduced model are identified and verified with the existing experimental research results. The yield surface equation of rock under true triaxial stress is obtained by data fitting, and the influence of intermediate principal stress on the creep model is discussed. The derived constitutive model can well describe the characteristics of each stage of true triaxial creep behavior of rock under acid environment.

## 2. Establishment of Nonlinear True Triaxial Creep Model Considering Chemical Damage

### 2.1. Damage Analysis

Zhang [24] improved the Lemaitre strain equivalent principle by defining the base damage state of rock. When rock is subject to the action of external force, the damage level will expand. Take two damage states: for rock materials in acidic environment, the effective stress of rock under the acid damage state acts on the strain caused by the acid; secondly, there is the stress joint damage state, which is equivalent to the effective stress of rock under the acid. Therefore, the stress joint damage state acts on the strain under the acid damage state:(1)$$\varepsilon =\frac{\sigma^{1}}{E^{2}}=\frac{\sigma^{2}}{E^{1}}$$
where: $\sigma^{1}$ and $E^{1}$ are the effective stress and elastic modulus of rock in acid damage state, respectively, $\sigma^{2}$ and $E^{2}$ are the effective stress and elastic modulus of rock under the condition of acid damage and stress damage, respectively.

Therefore, the next derivation process can be carried out:

Firstly, according to Lemaitre strain equivalence principle, it is obtained that:(2)$$E^{1}=E_{0}(1-D_{C})$$
where: *Dc* is the chemical damage caused by chemical corrosion, and *E*_0_ is the initial elastic modulus of rock.

Then, according to the improved Lemaitre strain equivalence principle, it is obtained that:(3)$$\sigma =E^{1}\varepsilon (1-D_{m}^{})$$
where: *D_m_* is the stress damage caused by stress loading.

Substitute Equation (2) into Equation (3) to obtain:(4)$$\sigma =E_{0}\varepsilon (1-D_{c}^{})(1-D_{m}^{})$$

According to the Equation (4), we can obtain:(5)$$\left(1-D_{m}\right)\left(1-D_{c}\right)=1-D_{cm}$$

So far, we obtain the expression of *D_s_* as follows:(6)$$D_{s}=1-\left(1-D_{m}\right)\left(1-D_{c}\right)$$
where: *D_s_* is the total damage caused by chemical corrosion and stress loading.

#### 2.1.1. Chemical Damage Variable

When the engineering rock was in an acidic environment, on the one hand, the rock was subjected to a load to form stress damage. On the other hand, the chemical composition in the acidic environment reacted with the rock, which caused the cement inside the rock to dissolve, so that the pores increased, the micro-cracks extended, and chemical damage was formed. The development and penetration of micro-cracks in the rock reflected the damage of the rock. Macroscopically, it showed the decrease in the elastic modulus. Therefore, according to damage mechanics, the damage degree expression of chemical corrosion on rock is obtained:(7)$$D_{c}=1-E_{C}/E_{0}$$
where: *E_C_* is the elastic modulus of rock in the environment at pH = C.

#### 2.1.2. Stress Damage Variable

When the stress on the rock is greater than its long-term strength σs, we believe that the rock will suffer from stress damage. According to the Kachanov creep damage law [25], we can obtain:(8)$${\dot{D}}_{m}=A{\left(\frac{\sigma }{1-D_{m}}\right)}^{N}$$
where *A* and *N* are the parameters related to the material, and *D_m_* is the stress damage variable.

According to Formula (8) and boundary conditions *D_m_* = 1, the evolution equation of the damage variable and creep time of rock during accelerated creep process is obtained as:(9)$$D_{m}=1-{\left(1-\frac{t}{t_{F}}\right)}^{\frac{1}{1+N}}$$

#### 2.1.3. Comprehensive Damage Variables under Chemical Corrosion and Stress

The damage caused by chemical corrosion exists throughout the creep process, whereas the stress damage occurs when the stress on the rock is greater than its long-term strength *σ*_s_. Combining Equations (6), (7), and (9), the comprehensive damage variables of rocks under the action of acid corrosion and stress can be obtained:(10)$$D=\left\{\begin{array}{l} 1-\frac{E_{C}}{E_{0}}\;\sigma \leq \sigma_{s}\\ 1-{\left(1-\frac{t}{t_{F}}\right)}^{\frac{1}{1+N}}+1-\frac{E_{C}}{E_{0}}-(1-\frac{E_{C}}{E_{0}})\left[1-{\left(1-\frac{t}{t_{F}}\right)}^{\frac{1}{1+N}}\right]\;\sigma >\sigma_{s} \end{array}\right.$$

### 2.2. Establishment of True Triaxial Creep Model

In the rock creep test, when the initial stress level is less than the long-term strength of the rock, the rock sample will first generate an instantaneous elastic strain during the loading process, since the loading time at this stage accounts for a very small part of the entire creep time. It can be considered that the elastic strain is completed instantaneously. The constitutive relationship at this stage can be described as an elastic body, and considering the true triaxial stress of the rock in the actual situation, the constitutive equation is:(11)$$\varepsilon_{ij}^{e}=\frac{1}{2G_{1}}S_{ij}+\frac{1}{3K_{1}}\delta_{ij}\sigma_{m}$$
where $\varepsilon_{ij}^{e}$ is the elastic strain under true triaxial stress, *S_ij_* is the deviatoric stress tensor, *δ_ij_σ_m_* is the spherical stress tensor, *G*_1_ is the shear modulus in the elastic strain stage, and *K*_1_ is the bulk modulus in the elastic strain stage.

Continuing to load, the rock enters the decay creep stage, and the creep curve at this stage has obvious nonlinear characteristics; therefore, it is difficult to accurately describe its characteristics by using the classical element model. In this paper, the viscosity coefficient of the sticky pot element in the traditional Kelvin model is modified by introducing the time–function relationship of the damage variable. That is, it is assumed that the viscosity coefficient has an exponential function relationship with the creep time during the rock creep process, so as to construct a nonlinear Kelvin model, and its three-dimensional differential constitutive equation satisfies:(12)$$\sigma_{ij}=G_{2}\varepsilon^{nve}+\eta_{2}e^{-\lambda t}{\dot{\varepsilon }}^{nve}$$
where *σ_ij_* is the stress, *G*_2_, *η*_2_ are the shear modulus and viscosity coefficient of the rock in the nonlinear viscoelastic stage, $\varepsilon^{nve}$ is the nonlinear viscoelastic strain, ${\dot{\varepsilon }}^{nve}$ is the first derivative of the nonlinear viscoelastic strain with time, λ is an undetermined constant, and *t* is creep time.

Separating the variables of the above differential equation and integrating with the initial conditions $t=0\;,{\dot{\varepsilon }}^{nve}=0$, the creep equation of the nonlinear Kelvin body can be obtained as:(13)$$\varepsilon^{nve}=\frac{S_{ij}}{2G_{2}}\left\{1-\exp\left[-\frac{G_{2}(e^{\lambda t}-1)}{\lambda \eta_{2}}\right]\right\}$$

As the stress continues to be loaded, viscoelastic strain occurs in the rock. In this paper, the Kelvin model is used to represent this stage in the rheological process of the rock. The constitutive equation is:(14)$$\varepsilon_{ij}^{\mathrm{ve}}=\frac{1}{2G_{3}}\left[1-\exp\left(-\frac{G_{3}t}{\eta_{3}}\right)\right]S_{ij}$$
where: $\varepsilon_{ij}^{\mathrm{ve}}$ is the viscoelastic strain under true triaxial stress, *G*_3_ is the shear modulus in the viscoelastic strain stage, and *η*_3_ is the viscosity coefficient in the viscoelastic strain stage.

When the applied stress level exceeds the long-term strength of the rock, the rock enters the accelerated creep stage. In the case of three-dimensional stress, if the stress state exceeds the viscoplastic yield surface, viscoplastic strain will begin to occur. The rate of change of viscoplastic strain is represented by the limit stress flow law of Perzyna [26,27]:(15)$${\dot{\varepsilon }}_{ij}{}^{vp}=\frac{1}{\eta_{4}(t)}\langle F\rangle \frac{\partial Q}{\partial \sigma_{ij}}$$
where:(16)$$\langle F\rangle =\left\{\begin{array}{l} 0\;,\;(F\leq 0)\;\\ F,\;(F>0) \end{array}\right.$$
where *F* is the yield function of the rock, *Q* is the plastic potential function, and *η*_4_ is the time-dependent viscosity coefficient.

Based on Equations (9), (15), and (16), the three-dimensional damage constitutive equation in the accelerated creep state can be obtained [28]:(17)$$\varepsilon_{ij}{}^{vp}=\frac{Ft_{F}\left(N+1\right)}{\eta_{4}N}\left[1-{\left(1-\frac{t}{t_{F}}\right)}^{\frac{N}{1+N}}\right]\frac{\partial F}{\partial \sigma_{ij}}$$

The constitutive models of the four stages are combined, as shown in Figure 1.

According to the principle of the series-parallel superposition of the models, the creep equation of the viscoelastic–plastic damage creep model of the rock under the true triaxial stress state can be obtained:(18)$$\varepsilon_{ij}^{}(t)=\left\{\begin{array}{ll} \frac{1}{2G_{1}}S_{ij}+\frac{1}{3K_{1}}\delta_{ij}\sigma_{m}+\frac{S_{ij}}{2G_{2}}\left\{1-\exp\left[-\frac{G_{2}(e^{\lambda t}-1)}{\lambda \eta_{2}}\right]\right\} & \left(F\;<\;0,\overset{\cdot }{\varepsilon }=0\right)\\ \frac{1}{2G_{1}}S_{ij}+\frac{1}{3K_{1}}\delta_{ij}\sigma_{m}+\frac{S_{ij}}{2G_{2}}\left\{1-\exp\left[-\frac{G_{2}(e^{\lambda t}-1)}{\lambda \eta_{2}}\right]\right\}+\frac{1}{2G_{3}}\left[1-\exp\left(-\frac{G_{3}t}{\eta_{3}}\right)\right]S_{ij} & \left(F\;<\;0,\overset{\cdot }{\varepsilon }>0\right)\\ \frac{1}{2G_{1}}S_{ij}+\frac{1}{3K_{1}}\delta_{ij}\sigma_{m}+\frac{S_{ij}}{2G_{2}}\left\{1-\exp\left[-\frac{G_{2}(e^{\lambda t}-1)}{\lambda \eta_{2}}\right]\right\}+\frac{1}{2G_{3}}\left[1-\exp\left(-\frac{G_{3}t}{\eta_{3}}\right)\right]S_{ij}\\ \;\;+\frac{Ft_{F}\left(N+1\right)}{\eta_{4}N}\left[1-{\left(1-\frac{t}{t_{F}}\right)}^{\frac{N}{1+N}}\right]\frac{\partial F}{\partial \sigma_{ij}} & \left(F\;\geq \;0\right) \end{array}\right.$$

In a true triaxial stress environment, the rock is stressed in three directions. For true triaxial rock creep tests, there are:(19)$$\left.\begin{array}{l} \sigma_{1}>\sigma_{2}>\sigma_{3}\\ I_{1}=\sigma_{1}+\sigma_{2}+\sigma_{3}\\ \sigma_{m}=\frac{\sigma_{1}+\sigma_{2}+\sigma_{3}}{3}\\ S_{11}=\sigma_{1}-\sigma_{m}=\frac{2\sigma_{1}-\sigma_{2}-\sigma_{3}}{3} \end{array}\right\}$$

In the process of triaxial compression, the failure mode of rock is mainly the compression shear failure sliding along the failure surface. Due to the formation and further development of cracks in the rock and the expansion of internal pores, the viscosity coefficient of the rock will be affected. Therefore, considering the influence of damage on shear modulus and viscosity coefficient, ignoring its influence on bulk modulus [29], there are:(20)$$\begin{array}{l} G\left(t\right)=G\left(1-D\right)\\ \eta \left(t\right)=\eta \left(1-D\right) \end{array}$$

Combining Equations (18)–(20), the creep equation of the viscoelastic-plastic chemical damage creep model of rock under true triaxial stress state can be obtained as:(21)$$\varepsilon_{11}(t)=\left\{\begin{array}{ll} \frac{2\sigma_{1}-\sigma_{2}-\sigma_{3}}{6G_{1}\left(1-D\right)}+\frac{\sigma_{1}+\sigma_{2}+\sigma_{3}}{9K_{1}}+\frac{2\sigma_{1}-\sigma_{2}-\sigma_{3}}{6G_{2}\left(1-D\right)}\left\{1-\exp\left[-\frac{G_{2}(e^{\lambda t}-1)}{\lambda \eta_{2}}\right]\right\} & \left(F\;<\;0,\overset{\cdot }{\varepsilon }=0\right)\\ \frac{2\sigma_{1}-\sigma_{2}-\sigma_{3}}{6G_{1}\left(1-D\right)}+\frac{\sigma_{1}+\sigma_{2}+\sigma_{3}}{9K_{1}}+\frac{2\sigma_{1}-\sigma_{2}-\sigma_{3}}{6G_{2}\left(1-D\right)}\left\{1-\exp\left[-\frac{G_{2}(e^{\lambda t}-1)}{\lambda \eta_{2}}\right]\right\}+\frac{2\sigma_{1}-\sigma_{2}-\sigma_{3}}{6G_{3}\left(1-D\right)}\left[1-\exp\left(-\frac{G_{3}t}{\eta_{3}}\right)\right] & \left(F\;<\;0,\overset{\cdot }{\varepsilon }>0\right)\\ \frac{2\sigma_{1}-\sigma_{2}-\sigma_{3}}{6G_{1}\left(1-D\right)}+\frac{\sigma_{1}+\sigma_{2}+\sigma_{3}}{9K_{1}}+\frac{2\sigma_{1}-\sigma_{2}-\sigma_{3}}{6G_{2}\left(1-D\right)}\left\{1-\exp\left[-\frac{G_{2}(e^{\lambda t}-1)}{\lambda \eta_{2}}\right]\right\}+\frac{2\sigma_{1}-\sigma_{2}-\sigma_{3}}{6G_{3}\left(1-D\right)}\left[1-\exp\left(-\frac{G_{3}t}{\eta_{3}}\right)\right]\\ \;\;+\frac{Ft_{F}\left(N+1\right)}{\eta_{4}N\left(1-D\right)}\left[1-{\left(1-\frac{t}{t_{F}}\right)}^{\frac{N}{1+N}}\right]\frac{\partial F}{\partial \sigma_{ij}} & \left(F\;\geq \;0\right) \end{array}\right.$$

### 2.3. Yield Surface Determination

Mogi [30] conducted true triaxial compression tests on various rocks. The test results showed that the existence of intermediate principal stress had an important influence on the strength of rock failure. Based on the von Mises criterion, Mogi proposed an octahedral strength stress criterion considering the intermediate principal stress. Al-Ajmi [31] analyzed multiple sets of rock true triaxial compression test data and found that the linear criterion proposed by Mogi was suitable for the true triaxial compression test of rock; this criterion is called the Mogi–Coulomb criterion, and its expression is as follows:(22)$$\tau_{\mathrm{oct}}=\delta_{1}+\delta_{2}\sigma_{m,2}$$
where:(23)$$\tau_{\mathrm{oct}}=\frac{1}{3}\sqrt{{\left(\sigma_{1}-\sigma_{2}\right)}^{2}+{\left(\sigma_{2}-\sigma_{3}\right)}^{2}+{\left(\sigma_{1}-\sigma_{3}\right)}^{2}}$$
(24)$$\sigma_{m,2}=\frac{\sigma_{1}+\sigma_{3}}{2}$$
where *τ*_oct_ is the octahedral stress, *σ*_*m*,2_ is the effective normal stress, and $\delta_{1}\;,\delta_{2}$ are experimental constants, which can be obtained by data fitting.

When the stress state of the rock reaches its stress yield surface, the rock will have obvious time-dependent behavior. Based on the Mogi–Coulomb criterion under three-dimensional stress, the damage stress yield surface of the rock in three-dimensional space is obtained as:(25)$$F=\delta_{2}\frac{I_{1}}{3}+\delta_{2}\sqrt{\frac{J_{2}}{3}}\left[\sin\left(\theta +\frac{2\pi }{3}\right)+\sin\left(\theta -\frac{2\pi }{3}\right)\right]-\sqrt{\frac{2J_{2}}{3}}+\delta_{1}$$
where:(26)$$\tan\theta =\frac{1}{\sqrt{3}}\frac{2\sigma_{1}-\sigma_{2}-\sigma_{3}}{\sigma_{1}-\sigma_{2}}$$
(27)$$I_{1}=\sigma_{1}+\sigma_{2}+\sigma_{3}$$
(28)$$J_{2}=\frac{\left[{\left(\sigma_{1}-\sigma_{2}\right)}^{2}+{\left(\sigma_{2}-\sigma_{3}\right)}^{2}+{\left(\sigma_{1}-\sigma_{3}\right)}^{2}\right]}{6}$$
where: *I*_1_ is the first invariant of stress, *J*_2_ is the second invariant of deviatoric stress, and θ is the Lode angle of stress.

## 3. Creep Model Parameter Identification and Model Verification

### 3.1. Parameter Identification

Due to the lack of laboratory creep test data considering the simultaneous action of acid corrosion and true triaxial stress, the conventional triaxial rock creep test after acid solution corrosion and the rock creep test under true triaxial stress were used for parameter identification and model verification.

Wang [21] conducted a conventional triaxial creep test of sandstone after acid corrosion. From this experiment, we observed that the elastic modulus of rock decreased obviously after the action of different pH values; this is the embodiment of chemical damage. We take the experimental data when the confining pressure is 6 MPa and pH = 3 to verify, the stress and strain are shown in the Figure 2 below:

According to Equation (7), we can obtain the value of the *Dc* when pH = 3 is 0.312. Wang conducted a conventional triaxial creep test of rock, which is a special case of *σ*_2_ = *σ*_3_. Therefore, the creep constitutive model deduced in this paper can be greatly simplified. Taking the long-term strength *σ*_s_ and strain rate $\overset{\cdot }{\varepsilon }$ during the rock creep process as the judgment conditions for each stage of the constitutive equation, the models that need to be determined are *G*_1_, *K*_1_, λ, *G*_2_, *η*_2_, *G*_3_, *η*_3_, *η*_4_, and *N*. The creep failure time *t*_F_ of the rock can be determined according to the test data. *G*_1_, *K*_1_ are determined according to the elastic modulus *E* and Poisson’s ratio *v* obtained from the conventional triaxial compression test of the rock under the same confining pressure, namely:(29)$$G_{1}=\frac{E}{2(1+\nu )},K_{1}=\frac{E}{3(1-2\nu )}$$

When in Formula (21) t→∞, we obtain:(30)$$\varepsilon_{11}^{t}(t\to \infty )=\frac{\sigma_{1}-\sigma_{3}}{3G_{1}(1-D)}+\frac{\sigma_{1}+2\sigma_{3}}{9K_{1}(1-D)}+\frac{\sigma_{1}-\sigma_{3}}{3G_{2}(1-D)}$$

Subtract the first Formula of (21) from Formula (30), and take the logarithm of both sides to obtain:(31)$$\ln\left[\varepsilon_{11}^{t}(t\to \infty )-\varepsilon_{11}^{t}\right]=\ln\frac{\sigma_{1}-\sigma_{3}}{3G_{2}(1-D)}-\frac{G_{2}(e^{\lambda t}-1)}{\lambda \eta_{2}}$$

Then order:$$\left\{\begin{array}{l} y(t_{i})=\ln\left[\varepsilon_{11}^{t}(t\to \infty )-\varepsilon_{11}^{t}\right],a=-\frac{G_{2}}{\lambda \eta_{2}},\\ b=e^{\lambda },c=\frac{G_{2}}{\lambda \eta_{2}}+\ln\frac{\sigma_{1}-\sigma_{3}}{3G_{2}} \end{array}\right.$$

The Formula (31) becomes:(32)$$y(t_{i}){=\mathrm{ab}}^{t}+c$$

Finally, the nonlinear fitting analysis is performed on the experimental data, and the fitting parameters a, b, and c can be obtained, then $\lambda ,G_{2},\eta_{2}$ are:(33)$$\left.\begin{array}{l} \lambda =\ln b\\ G_{2}=\frac{\sigma_{1}-\sigma_{3}}{3e^{c+a}(1-D)}\\ \eta_{2}=-\frac{\sigma_{1}-\sigma_{3}}{3{\mathrm{ae}}^{c+a}\ln b} \end{array}\right\}$$

The creep failure time *t*_F_ of the rock can be determined according to the rock creep test, and the remaining model parameters *G*_3_, *η*_3_, *η*_4_, and *N* are determined by inversion based on the principle of least squares. The Levenberg–Marquardt algorithm that comes with Origin software is used for nonlinear regression analysis to identify the parameters of the rock creep model. The algorithm improves the least squares method and introduces a damping factor d to avoid iterative non-convergence [32]. Table 1 shows the rock creep model parameters considering chemical damage. Figure 3 shows the comparison between the rock creep test curve and the constitutive curve considering chemical damage.

Zhao [17] carried out the creep test of rock under true triaxial stress with Jinping marble, and obtained the mechanical behavior of rock creep under different intermediate principal stresses, which can be regarded as a special case where the chemical damage factor is 0. Therefore, it is used to verify the applicability of the constitutive model derived in this paper under the true triaxial stress state. The creep data under the action of true triaxial stress are shown in Table 2 below. According to the data, we can obtain the linear relationship between the octahedral shear stress *τ*oct and the average normal stress *σ*_*m*,2_, and thus obtain the size of the constant in formula (22), as shown in Figure 4, *C*_1_ = 0.608, *C*_2_ = 17.2, and substituting it into formula (25) to obtain the expression of the yield surface.

The determination of the parameters of the constitutive model under the action of true triaxial stress is consistent with the above process. In the case of three-dimensional stress, this paper selects rock creep test data under different intermediate principal stresses and rock creep data under different maximum principal stresses, respectively, for verification. The creep parameters of marble under the true triaxial stress state were obtained as shown in Table 3. The experiments conducted by Zhao did not consider the chemical damage of the rock. Therefore, for the rock creep equation derived in this paper considering chemical damage, the chemical damage factor *D_c_* is 0. Combined with the yield surface constants *C*_1_, *C*_2_ obtained above and the determined parameters of the constitutive model, the comparison between the test curve and the model curve of the rock under the action of true triaxial stress is shown in Figure 5.

### 3.2. Constitutive Model Verification

Comparing the experimental and theoretical curves in Figure 3 and Figure 5, we can conclude that the true triaxial creep model curve under the action of rock acid corrosion deduced in this paper is in agreement with the experimental curve, which not only reflects the effect of chemical damage on rock creep characteristics, but also reflects the creep behavior of rock under the action of true triaxial stress, including elastic stage, decay stage, constant velocity stage, and acceleration stage of rock creep. Different intermediate principal stresses will have an important influence on the creep behavior of rocks. It can be seen that when the intermediate principal stress is too large or too small, the conformity between the constitutive curve and the test curve will decrease to a certain extent. At the same time, it can be seen that when the rock creep is in the acceleration stage, the data of the constitutive curve and the test curve often have a certain deviation, and the deviation is about 10%. This is because the stress on the rock at this stage is greater than its long-term strength, and the cracks generated on the surface and inside of the rock begin to extend and penetrate, resulting in uneven distribution of stress inside the rock at this stage. On the whole, the true triaxial creep model considering chemical corrosion established in this paper can accurately reflect the creep behavior of rock, which verifies the rationality of the model establishment and the correctness of the parameter verification.

## 4. Influence of Model Parameters on Intermediate Principal Stress

The rock creep constitutive model established in this paper considers the effect of true triaxial stress. Therefore, it is necessary to explore the influence of different intermediate principal stresses on the model parameters. The creep model parameters of rock under different intermediate principal stresses are shown in Figure 6 below:

As can be seen from Figure 6, the value of the shear modulus *G*_1,_ as an elastic parameter, increases gradually with the increase in the intermediate principal stress. This shows that with the increase in the intermediate principal stress, the original cracks, holes, and other micro-defects in the rock are closed, so that the rock sample is hardened to a certain extent. The viscosity coefficients *η*_2_, *η*_3_ are used to describe the ability of rock samples to resist deformation, and their values show an upward trend with the increase in the intermediate principal stress, which means that the increase in the intermediate principal stress inhibits the time-dependent deformation of the rock, so at the macro level, it is manifested as a decrease in the creep rate; meanwhile, the material parameter *λ* shows a decreasing trend with the increase in the intermediate principal stress.

## 5. Conclusions

Based on damage mechanics, chemical damage is connected with elastic modulus; thus, the damage relations considering creep stress damage and chemical damage are obtained. The elastic body, nonlinear Kelvin body, linear Kelvin body, and viscoelastic–plastic body are connected in series, and the actual situation under the action of true triaxial stress is considered at the same time. Therefore, a damage creep constitutive model considering the coupling of rock acid corrosion and true triaxial stress is established, which is verified with the experimental data, and the following conclusions are obtained:

1.The constitutive model established in this paper can accurately reflect the creep characteristics of rock under acid corrosion and true triaxial stress state, and the model’s fitting degree is more than 90%, which verifies its rationality and the correctness of parameter determination.2.The creep model cannot only accurately describe the creep curve characteristics of the rock in the transient elastic strain stage and the constant velocity creep stage, but also can describe the nonlinear characteristics of the creep curve in the attenuation creep stage and the accelerated creep stage.3.The influence of different intermediate principal stresses on rock parameters is analyzed. The value of shear modulus *G*_1_ increases with the increase in intermediate principal stress, and the viscosity coefficients *η*_2_, *η*_3_ increase with the increase in intermediate principal stress. The material parameters show a decreasing trend with the increase in the intermediate principal stress.

## Figures and Tables

**Figure 1 materials-15-07590-f001:**
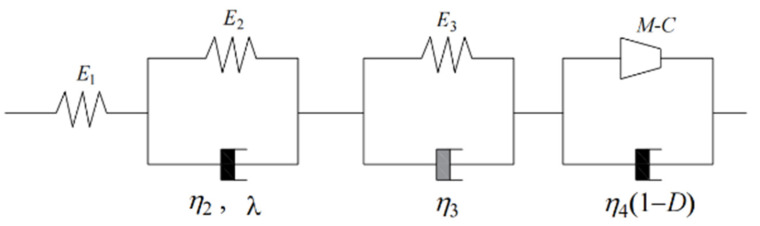
Nonlinear viscoelastic plastic damage creep mechanical model.

**Figure 2 materials-15-07590-f002:**
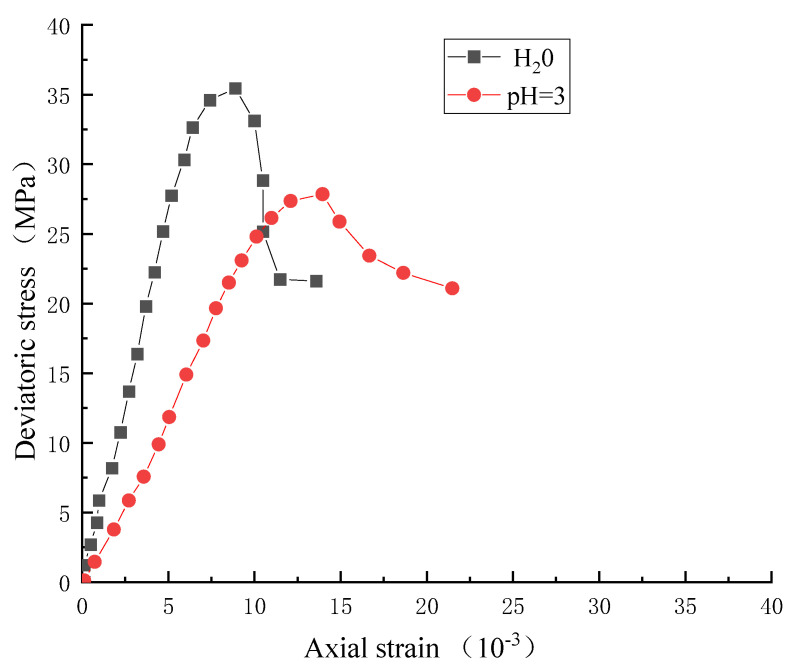
Experimental stress-strain curve.

**Figure 3 materials-15-07590-f003:**
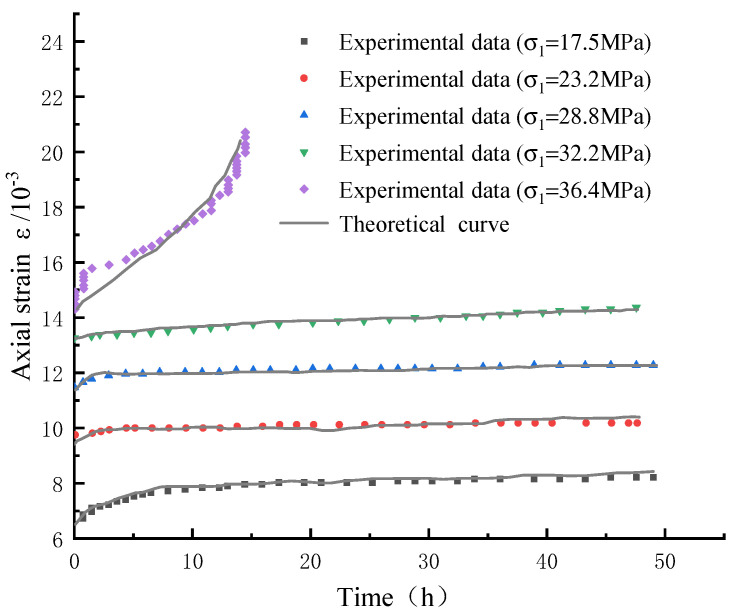
Comparison of rock creep experimental curve and constitutive curve considering chemical damage.

**Figure 4 materials-15-07590-f004:**
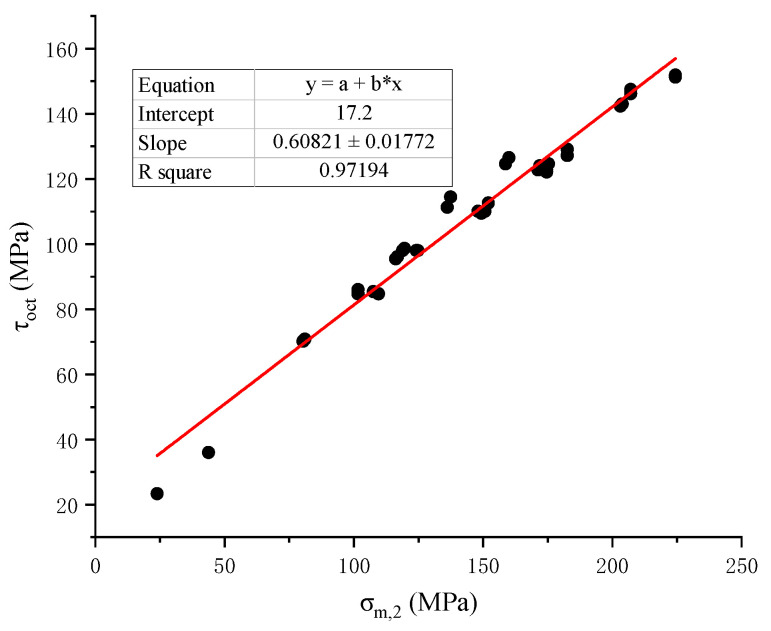
Linear relationship between octahedral shear stress and mean normal stress.

**Figure 5 materials-15-07590-f005:**
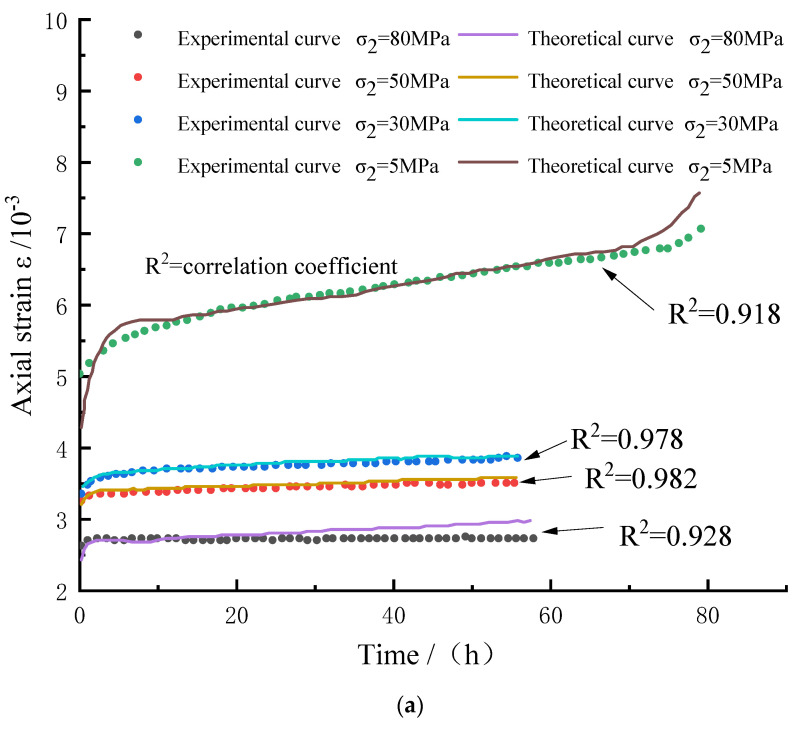
Comparison between experimental and model curves of marble under true triaxial stress. (**a**) *σ*_3_ = 5 MPa, *σ*_1_–*σ*_3_ = 220 MPa; (**b**) *σ*_3_ = 5 MPa, *σ*_2_ = 80 MPa.

**Figure 6 materials-15-07590-f006:**
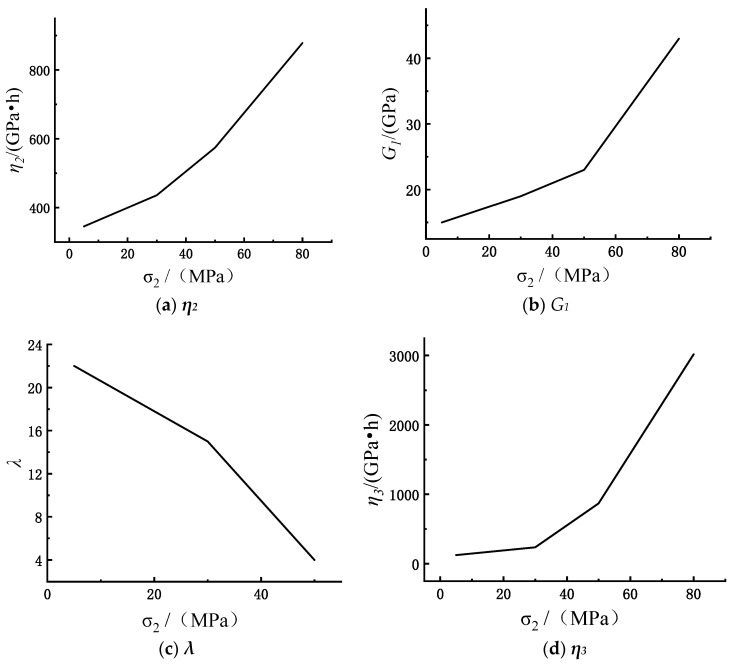
Effects of different intermediate principal stresses on model parameters.

**Table 1 materials-15-07590-t001:** Sandstone triaxial creep model parameters considering chemical damage (initial pH = 3).

*σ*_1_–*σ*_3_	*K*_1_/GPa	*G*_1_/GPa	*G*_2_/GPa	*η*_2_/(GPa·h)	*λ*	*G*_3_/GPa	*η*_3_/(GPa·h)	*η*_4_/(GPa·h)	*t_F_*	*N*
11.5	1.81	3.21	1.27	0.29	0.081	-	-	-	-	-
17.2	1.82	3.22	2.60	0.15	0.027	-	-	-	-	-
22.8	1.84	3.12	1.78	0.15	0.037			-	-	-
22.6	1.83	3.24	1.78	0.007	−0.066	24.5	1.32	-	-	-
30.4	1.87	3.11	0.05	10.82	0.032	34.6	1.66	23.0	198.4	0.8

**Table 2 materials-15-07590-t002:** Rock peak stress under true triaxial stress.

*σ*_3_ (MPa)	*σ*_2_ (MPa)	*σ*_p_ (MPa)
0	0	190
0	60	253
5	5	225
5	30	265
5	50	281
5	100	306
10	50	325
15	15	285
15	30	289
20	50	356
30	30	348
30	40	373
30	50	384
30	80	409
30	105	413
30	120	425
30	150	441
40	50	415
40	100	461
40	200	511

**Table 3 materials-15-07590-t003:** Parameters of the marble creep model under true triaxial stress.

*σ*_1_ (MPa)	*σ*_2_ (MPa)	*σ*_3_ (MPa)	*K*_1_ (GPa)	*G*_1_ (GPa)	*G*_2_ (GPa)	*η*_2_ (GPa.h)	*λ*	*G*_3_ (GPa)	*η*_3_ (GPa.h)	*η*_4_ (GPa.h)	*t_F_*	*N*
225	5	5	94	15	131	345	−0.54	45	123	632	79.4	22
225	30	5	87	19	242	436	−0.62	123	234			
225	50	5	93	23	423	574	−0.67	435	867			
225	80	5	85	43	675	878	−0.86	845	3018			
245	80	5	98	34	1209	1245	−0.86	1245	5632			
265	80	5	79	45	1634	1865	−0.82	1431	2324			
285	80	5	76	65	2313	2344	−0.78	2562	2345			
305	80	5	88	62	1543	1645	−0.98	865	978			
325	80	5	96	67	890	1099	−0.79	897	790	890	2.11	5

## Data Availability

Data sharing is not applicable to this article.

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
