# Peer review of "A Chemical Damage Creep Model of Rock Considering the Influence of Triaxial Stress"

_materials, 2022, doi:10.3390/ma15217590_

Round 1

Reviewer 1 Report

Dear Authors,

The paper presents a model of rock creep under the influence of a specific environment. A constitutive creep model was established taking into account the coupling of acid corrosion and the actual triaxial stress. The analysis of the results showed that the obtained constitutive model may well reflect the creep properties of the rock under the influence of acid corrosion. The authors presented and described the methodology of the work well, presented the model and compared it experimentally.The manuscript is well written, contains all the important issues, and the division of works is correct and legible, although I pay attention to chapter 4. The conclusions obtained are compact and to the point, but they could be preceded by at least a short introductory sentence. the choice of literature is also appropriate, although it is largely limited to works from one part of the world. I can suggest some rock studies carried out in the region of Eastern Poland: DOI: 10.1111 / j.1475-4754.2009.00507.x (lines 27-30).

In general, I evaluate the work well, although some editing errors should be corrected (for example lines 168, 181, 273) and the entry of the list of references should be adapted to the journal template.

In addition, I have the following comments on the manuscript:

Figure 1 should be slightly enlarged, the symbols marked in the figure are illegible.

Correct the description of the timeline in Figure 2 and 3 and 5 are divergent can be written as - Time (h)

Chapter 4: please do not end the chapter with a drawing, maybe the description should be modified and some text should be moved under Figure 6.

In my opinion, the manuscript may be recommended for publication only after the above-mentioned comments have been introduced.

Author Response

Dear Reviewer:

Thank you for the comments concerning our manuscript entitled “A chemical damage creep model of rock considering the influence of triaxial stress” (Title has been modified) (ID: materials-1970237). Those comments are all valuable and very helpful for revising and improving our paper, as well as crucial guiding significance to our research. We have studied comments carefully and made corrections, which we hope meet with approval.

Revised portions were marked in the manuscript. The corrections in the manuscript and the responses to the reviewer’s comments were as follows:

Responds to the reviewer’s comments:

1.“The conclusions obtained are compact and to the point, but they could be preceded by at least a short introductory sentence”

Thanks for pointing out this question, we have supplemented this part. Modifications have been made in the manuscript at line 313

  1. the choice of literature is also appropriate, although it is largely limited to works from one part of the world. I can suggest some rock studies carried out in the region of Eastern Poland: DOI: 10.1111 / j.1475-4754.2009.00507.x (lines 27-30).

Thanks for pointing out this question, we have supplemented this part. Modifications have been made in the manuscript at line 22

3.Figure 1 should be slightly enlarged, the symbols marked in the figure are illegible.

Thanks for pointing out this question, we have supplemented this part. odifications have been made in the manuscript at line 166

4.Correct the description of the timeline in Figure 2 and 3 and 5 are divergent can be written as - Time (h)

Thanks for pointing out this question, The abscissa here is time and a positive value. The creep curve changes with the increase of loading time.

  1. Chapter 4: please do not end the chapter with a drawing, maybe the description should be modified and some text should be moved under Figure 6.

Thanks for pointing out this question, we have supplemented this part. Modifications have been made in the manuscript at line 301

Finally, we corrected some spelling errors and the format of references in the paper, and again thank the reviewers for your valuable comments.

Reviewer 2 Report

The paper studies the rock creep behavior under acid corrosion and true triaxial stress state.

The paper looks good after few additions as suggested below:

1.Adding a discussion / result for validation of the constitutive model to a real project. The verification of the model with theoretical results are not enough and need further justification regarding its applicability to a real case scenario.

2.Font size in Figure 5 is not readable.

3.Remove the double numbering in the  conclusion section

4.The reference section need to be modified as per the journal standard.

Author Response

Dear Reviewer:

Thank you for the comments concerning our manuscript entitled “A chemical damage creep model of rock considering the influence of triaxial stress” (Title has been modified) (ID: materials-1970237). Those comments are all valuable and very helpful for revising and improving our paper, as well as crucial guiding significance to our research. We have studied comments carefully and made corrections, which we hope meet with approval.

Revised portions were marked in the manuscript. The corrections in the manuscript and the responses to the reviewer’s comments were as follows:

Responds to the reviewer’s comments:
1.Adding a discussion result for validation of the constitutive model to a real project. The verification of the model with theoretical results are not enough and need further justification regarding its applicability to a real case scenario.

Thanks for pointing out this question, the theoretical model in this paper is verified by actual experimental data, comparing the experimental and theoretical curves in Figures 3 and 5, it can be seen that the true triaxial creep model curve under the action of rock acid corrosion deduced in this paper is in good agreement with the experimental curve, which can not only reflect the effect of chemical damage on rock creep characteristics. At the same time, it can reflect the creep behavior of rock under the action of true triaxial stress.

2.Font size in Figure 5 is not readable.

Thanks for pointing out this question, we have supplemented this part, Modifications have been made in the manuscript at line 276

3.Remove the double numbering in the conclusion section

Thanks for pointing out this question, we have supplemented this part, Modifications have been made in the manuscript at line 313

4.The reference section need to be modified as per the journal standard.

Thanks for pointing out this question, we have supplemented this part, Modifications have been made in the manuscript at line 332

Finally, we corrected some spelling errors and the format of references in the paper, and again thank the reviewers for your valuable comments.

Reviewer 3 Report

The paper describes A True Triaxial Creep Constitutive Model for Rock Considering Hydrochemical Damage. All theoretical settings of the model itself are given. However, there is no description of how to solve theoretical assumptions or equations. How was this problem modeled? Whether a new numerical model was developed or an existing one was used with the modification of certain parameters. I ask the authors to describe this problem in more detail.

Experimental research was carried out in the paper. You should definitely insert photos of the test and describe the experimental research in more detail.

The conclusions should be expanded a little and supported with concrete values.

Author Response

Dear Reviewer:

Thank you for the comments concerning our manuscript entitled “A chemical damage creep model of rock considering the influence of triaxial stress” (Title has been modified) (ID: materials-1970237). Those comments are all valuable and very helpful for revising and improving our paper, as well as crucial guiding significance to our research. We have studied comments carefully and made corrections, which we hope meet with approval.

Revised portions were marked in the manuscript. The corrections in the manuscript and the responses to the reviewer’s comments were as follows:

Responds to the reviewer’s comments:
1.The paper describes A True Triaxial Creep Constitutive Model for Rock Considering Hydrochemical Damage. All theoretical settings of the model itself are given. However, there is no description of how to solve theoretical assumptions or equations. How was this problem modeled? Whether a new numerical model was developed or an existing one was used with the modification of certain parameters. I ask the authors to describe this problem in more detail.

Thanks for pointing out this question. In this paper, a creep constitutive model considering triaxial stress and chemical damage is established and verified with the experimental data of previous scholars. The parameters of the deduced model are identified and verified with the existed experimental research results. The yield surface equation of rock under true triaxial stress is obtained by data fitting, and the influence of intermediate principal stress on the creep model is discussed. The research in this paper is full and scientific.

2.Experimental research was carried out in the paper. You should definitely insert photos of the test and describe the experimental research in more detail.

Thanks for pointing out this question. There is no experimental study in this paper, and the data used are the experiments of previous scholars. This paper only uses their data to verify the creep model derived in this paper.

3.The conclusions should be expanded a little and supported with concrete values.

Thanks for pointing out this question. Thanks for pointing out this question, we have supplemented this part, Modifications have been made in the manuscript at line 313

Finally, we corrected some spelling errors and the format of references in the paper, and again thank the reviewers for your valuable comments.

Round 2

Reviewer 1 Report

Dear Authors,

Thanks for the revisions to the manuscript. After reading the text, I accept the article in its current form.

Thank you.

Reviewer 3 Report

The authors have corrected the paper according to the reviewers' requirements and the paper can be published.